# Piezoelectric Properties of 0-3 Composite Films Based on Novel Molecular Piezoelectric Material (ATHP)_2_PbBr_4_

**DOI:** 10.3390/ma15186378

**Published:** 2022-09-14

**Authors:** Xin Guo, Jialin Zhu, Xiaoping Zou, Junming Li, Jin Cheng, Chunqian Zhang, Yifei Wang, Xiaolan Wang, Hao Wang, Xinyao Chen, Weimin Wang, Mingkai Gu, Shixian Huang, Ruoxia Gui

**Affiliations:** 1Research Center for Sensor Technology, Beijing Key Laboratory for Sensor, Jianxiangqiao Campus, Beijing Information Science and Technology University, Beijing 100101, China; 2School of Science, Jianxiangqiao Campus, Beijing Information Science and Technology University, Beijing 100101, China; 3School of Automation, Jianxiangqiao Campus, Beijing Information Science and Technology University, Beijing 100101, China; 4Beijing Advanced Innovation Center for Materials Genome Engineering, Jianxiangqiao Campus, Beijing Information Science and Technology University, Beijing 100101, China

**Keywords:** 0-3 type piezoelectric composite film, ferroelectric polarization, piezoelectric response

## Abstract

Since their discovery, ferroelectric materials have shown excellent dielectric responses, pyroelectricity, piezoelectricity, electro-optical effects, nonlinear optical effects, etc. They are a class of functional materials with broad application prospects. Traditional pure inorganic piezoelectric materials have better piezoelectricity but higher rigidity; pure organic piezoelectric materials have better flexibility but havetoo small a piezoelectric coefficient. The material composite, on the other hand, can combine the advantages of both, so that it has both flexibility and a high piezoelectric coefficient. In this paper, a new molecular piezoelectric material (C_5_H_11_NO)_2_PbBr_4_ with a high Curie temperature Tc and a large piezoelectric voltage constant g_33_, referred to as (ATHP)_2_PbBr_4_, was used to prepare a 0-3 type piezoelectric composite film by compounding with an organic polymer material polyvinylidene fluoride (PVDF), and its ferroelectricity was investigated. The results show that the 0-3 type (ATHP)_2_PbBr_4_ piezoelectric composite film has good ferroelectricity and piezoelectricity, and the calculated piezoelectric voltage constant g_33_ after polarization is about 358.6 × 10^−3^ Vm/N, which is higher than that of PVDF material, and is important for the fabrication of high-performance piezoelectric sensors.

## 1. Introduction

With the development of science and technology, modern industries are beginning to demand integrated electronic components, and tiny-scale piezoelectric films, especially composite piezoelectric films combining piezoelectric ceramics and polymers, have received extensive attention from research scholars in various industries [1]. Piezoelectric ceramic materials have a larger piezoelectric coefficient compared to polymers, but they also have a higher elastic modulus and are therefore stiffer than polymers, which makes ceramic materials insensitive to small vibrations and more susceptible to stress damage [2]. Compared to piezoelectric ceramics and piezoelectric crystals, piezoelectric polymers have smaller piezoelectric stress constants (d_31_) than piezoelectric ceramics, yet piezoelectric polymers have much higher piezoelectric voltage constants (g_31_) than piezoelectric ceramics, which suggests that they are better sensor materials than piezoelectric ceramics [3]. It is easy to see that ceramic materials have satisfactory piezoelectricity and low dielectric loss; however, this rather increases the difficulty of device integration, as piezoelectric ceramic materials are quite fragile. In contrast, polymers have good mechanical and processing properties, as well as high breakdown strength, but a disadvantage of polymers is their low piezoelectric strain constants [2,4]. Therefore, it is valuable to obtain both good piezoelectric properties of ceramic materials and good mechanical strength and processing properties of polymers by compounding polymers with inorganic piezoelectric materials to become composite piezoelectric materials.

According to the connectivity of each material, piezoelectric composites can be divided into 10 basic types, namely, 0-0, 0-1, 0-2, 0-3, 1-1, 1-2, 1-3, 2-2, 2-3, and 3-3 types, and it is customary to use the former number to label the ceramic phase and the latter number to indicate the polymer phase. Among them, type 0-3 is the simplest form, which is formed by piezoelectric ceramic powder dispersed in a polymer matrix and has the characteristics of good flexibility and various processing shapes. Polyvinylidene fluoride (PVDF) is one of the main substrate materials used in composite functional materials. Xu [5] et al. prepared lead zirconate titanate (PZT)/polyvinylidene fluoride (PVDF) 0-3 type piezoelectric composites doped with polyaniline (PANI) by the hot-pressing method and characterized the polarization properties of the composites and also measured their piezoelectric properties. The results showed that the polarization of PZT could be carried out efficiently, and the piezoelectric coefficient increased and showed polar values at 8 vol%–10 vol% PANI by increasing the volume fraction of PANI in the composites. Luo [6] et al. used dopamine modification to combine calcium zirconate titanite (Ba_0.95_Ca_0.05_Zr_0.15_Ti_0.85_O_3_, BCZT) ceramic particles and PVDF to form 0-3 type composite flexible films and found that the dielectric constant increased and the dielectric loss decreased, both with increases in ceramic powder content, and the accuracy of the experiments was verified by combining simulation analysis with experiments. Wang [7] successfully prepared 0-3 type sodium potassium niobate (KNN)/polyvinylidene fluoride (PVDF) piezoelectric composite films and showed that KNN doping can disrupt the internal structure of the PVDF matrix, and the β-phase becomes a lower energy conformation when the defects in the PVDF matrix increase. In contrast, after the polarization of KNN/PVDF composite piezoelectric films, the α-phase in the PVDF matrix transforms into the β-phase, leading to an increase in the β-phase content. Brunengo [8] et al. incorporated up to 30% volume of lead zirconate titanate (PZT) particles into polyvinylidene fluoride (PVDF) by different techniques to explore the effect of different molding processes on PVDF polymorphism and molecular relaxation and the dielectric response of the composites, and it was found that the solution casting method resulted in a uniform dispersion of ceramic particles in PVDF, which led to a significant increase in the β-phase of PVDF, resulting in improved piezoelectric and dielectric properties. Yang [9] et al. also used dopamine as a surface modifier, and barium titanate and PVDF for mixing using a PDA modification strategy to improve the dispersion of inorganic materials and to reduce interfacial holes, defects, and cracks between the two components. The prepared nanogenerator exhibited a fast response of 61 ms and a significant piezoelectric output voltage of 9.3 V. Vinoth [10] et al. prepared electrospun pure polymer poly(vinylidene fluoride-co-hexafluoropropylene) (P(VDF-HFP)) fibrous membranes (esPFM) and electrospun nanocomposite polymer (P(VDF-HFP) + TiO_2_ nanofiber filler (2, 4, 6, and 8 wt%)) fibrous membranes (esNCPFMs) using the electrospinning technique. The electrospun nanocomposite polymer (PVDF-HFP + 6wt% TiO_2_ nanofiber filler) fiber membrane electrolyte was found to show the highest electrical conductivity of 1.87 × 10^−2^ S/cm at room temperature very well compared to the liquid electrolyte system. It can be a better electrolyte for developing high-efficiency dye-sensitized solar cell (DSSC) applications. Using the same approach, Vinoth Subramanian [11] et al. developed electrospun nanocomposite polymer blend poly(vinylidene fluoride-co-hexafluoropropylene) (PVDF-HFP)/poly(methyl methacrylate) (PMMA) films with x wt% of one-dimensional (1D) TiO_2_ nanofiber fillers dispersed on them (x = 0.0–0.8, step size 0.2). The corresponding dye-sensitized solar cells (DSSC) were also fabricated, which possessed a high electrical conductivity of 2.80 × 10^−2^ S/cm and a power conversion efficiency (PCE) of 8.08%. It was finally demonstrated that electrospun hybrid polymer nanocomposites (PVDF-HFP-PMMA-6 wt% TiO_2_ nanofiber filler) with LiI-based redox agent and tert-butyl pyridine (TBP) additives as polymer quasi-solid electrolyte nanofiber membranes could be better electrolytes for high-performance dye-sensitized solar cell applications.

As a useful complement to inorganic ferroelectrics and ferroelectric polymers, molecular ferroelectrics can match or exceed inorganic ferroelectrics in terms of properties such as large spontaneous polarization, high Curie temperature, and large piezoelectric response [12]. The molecular ferroelectric (C_5_H_12_NO)_2_PbBr_4_ (referred to as (ATHP)_2_PbBr_4_), which was discovered by a research team from Southeast University [13] in 2020, has excellent ferroelectricity and piezoelectricity. The g_33_ is 660.3 × 10^−3^ Vm/N, more than twice that of PVDF (g_33_=286.7 × 10^−3^ Vm/N), and far beyond that of conventional piezoelectric ceramics such as PZT. It also has a piezoelectric coefficient d_33_ of 76 pC/N and a Curie temperature Tc of 503 K (much higher than that of BTO at 393 K), which further guarantees that it can be adapted to a wider range of applications. In this paper, a new molecular ferroelectric (ATHP)_2_PbBr_4_ and an organic polymer, polyvinylidene fluoride (PVDF), were laminated for the first time to produce a 0-3 piezoelectric composite film. The aim is to obtain composite films with both excellent piezoelectric properties and good mechanical properties. It will show great potential for the next generation of smart piezoelectric sensor applications and become a strong contender in areas such as flexible devices, soft robotics, and biomedical devices.

## 2. Materials and Methods

Preparation of the instrument type and preparation of (ATHP)_2_PbBr_4_ powder as well as the cleaning procedure of ITO conductive substrates used in the experiments are detailed in the supporting literature and will not be repeated here.

### 2.1. Experimental Material

The experimental substrates used were ITO conductive surface glass substrate and flexible substrate PEN (polyethylene naphthalene dicarboxylate; manufacturer is South China Xiangcheng Technology Co. from Guangzhou, China), which could reach more than 85%; the chemicals used in this paper are shown in Table 1.

De-ionized water, anhydrous ethanol, isopropanol, and acetone were used for cleaning ITO conductive glass and flexible ITO substrate; lead bromide, 4-aminotetrahydropyran, and hydrobromic acid were used for preparing (ATHP)_2_PbBr_4_ powder; anhydrous dimethylformamide (DMF) was used as a solvent to dissolve polyvinylidene fluoride and (ATHP)_2_PbBr_4_ powder; and copper tape was used as the conductive electrode.

### 2.2. Preparation of 0-3 Type Piezoelectric Composite Film

The composite films were prepared as follows:(1)First,0.5 g of (ATHP)_2_PbBr_4_ powder and 2 g of PVDF powderwere weighed, and 2 mL and 8 mL of DMF were added dropwise to the two powders.(2)The PVDF was heated and stirred at 60 °C until it became a clarified colloid. The (ATHP)_2_PbBr_4_ precursor solution was heated and stirred and then transferred to an ultrasonic disperser for sonication and filtration of the solution.(3)The filtered (ATHP)_2_PbBr_4_ precursor solution was added to the already-stirred PVDF colloidal solution, the mixture was heated and stirred for 3 h, and then the stirrer was switched off for heating only and left to stand overnight.(4)Type 0-3 piezoelectric films were prepared using the spin-coating method, with the homogenizer speed adjusted to 500 rpm for 20 s in the first stage and 1000 rpm for 20 s in the second stage. Then 100 μL of Type 0-3 precursor solution was spin-coated onto the conductive surface and annealed at 90 °C for 30 min to produce Type 0-3 piezoelectric composite films.

The concentration of (ATHP)_2_PbBr_4_ powder in the final composite film was 0.25 g/mL.

### 2.3. Test Instruments and Methods

1. Scanning electron microscope (SEM): Model No. GeminiSEM 300, ZEISS, Jena, Germany.

2. X-ray diffraction (XRD): Model D8 Focus, manufacturer Bruker, Germany (Branch in Beijing China). This test instrument was provided by the Institute of Physical and Chemical Technology, Chinese Academy of Sciences. The test was performed using a Cu target with a wavelength of 1.5406 Å; the scan 2θ range was 5–50°,and the scan speed was set to 0.1 deg/s.

3. Differential scanning calorimeter (DSC): Model DSC8500, manufacturer PerkinElmer, USA (Branch in Shanghai China). This test instrument was provided by the National Center for Nanoscience. Nitrogen was used as the protective gas, and the scan rate was 30 °C/min at atmospheric pressure, and the DSC test temperature range was set between 120 °C and 240 °C.

4. Piezoelectric force microscope (PFM): Model Dimension Icon, manufacturer Bruker (Germany). This test instrument was provided by the Institute of Electrical Engineering, Chinese Academy of Sciences.

## 3. Results and Discussion

X-ray diffraction analysis of the previously prepared (ATHP)_2_PbBr_4_ powder was carried out at room temperature (Appendix A). The XRD patterns of the experimentally prepared (ATHP)_2_PbBr_4_ crystalline powder were compared with the standard comparison card for this substance. The comparison revealed that the peak positions of the characteristic peaks corresponding to the indices of each crystalline surface matched almost exactly with the XRD of the standard comparison card, which indicates a good reduction of the ferroelectric material from the literature [13]. Subsequent DSC experiments were performed on the (ATHP)_2_PbBr_4_ powder (Appendix A), and a reversible phase transition was measured at 220 °C/207 °C, with 220 °C being the Curie temperature of the material.

### 3.1. X-ray Energy Spectrum Analysis of 0-3 Type Piezoelectric Films

The structure of our prepared 0-3 piezoelectric thin film is shown schematicallyin Figure 1. After the 0-3 piezoelectric films were prepared by the homogenization method, the elements present and their contents were analyzed by X-ray energy spectrum analysis to further determine the material as well as to verify the accuracy of XRD. Figure 2a shows the energy spectrum surface scan area, which shows that the film surface is relatively homogeneous, with granular objects embedded in the film and fewer holes on the film surface. Figure 2b–f shows the distribution of EDS elements, Figure 2b shows the total map of all elements, and Figure 2c–f shows the distribution of these elements, Br, C, F, and Pb, respectively. Where F element is provided by PVDF material, Br and Pb are provided by (ATHP)_2_PbBr_4_ material. The locations where granular objects exist in the film are Br and Pb elements, and these two elements appear aggregated at the particle locations and correspond to each other, and in the locations where no particles exist, the content and distribution of these two elements are less, but they are still present and belong to the residual situation. From Figure 2e, the distribution of the F element is just complementary to that of Br and Pb. The part of the figure with less green in e is exactly the particle part, and the part with more is exactly the material containing the F element. Figure 2g shows the X-ray energy spectrum content, which is obtained by the surface scan of Figure 2a. The four main peaks present in the figure correspond to the four elements of C, F, Br, and Pb, respectively. The remaining small peaks should be detected in the conductive substrate because we set the acceleration voltage to 15 kV, and large acceleration voltages tend to pass through the film material and thus through the substrate. Therefore, the analysis of these figures shows that the particle part is the piezoelectric material (ATHP)_2_PbBr_4_, and the rest is composed of PVDF colloid, which happens to form a relatively uniform 0-3 piezoelectric film state, with the colloid wrapped around the piezoelectric ceramic material. Table 2 shows the weight and atomic percentages of the elements and their species contained in the energy surface scan total spectrum, where the atomic percentages of Br and Pb are 2.42 and 0.66, respectively. Based on the chemical formula of (ATHP)_2_PbBr_4_, the ratio of Pb to Br is 1:4, and the ratio of Pb to Br is calculated from the table to be approximately 1:4 as well, which further proves the accuracy of our preparation materials.

### 3.2. XRD of 0-3 Type Piezoelectric Films

Figure 3 shows XRD plots of 0-3 piezoelectric films at an annealing temperature of 90 °C and a rotational speed of 500–1000 rpm. The results of previous XRD characterization of (ATHP)_2_PbBr_4_ powders were compared with them. Some of the main peaks of (ATHP)_2_PbBr_4_ powder are indicated in the figure, such as at 6.5° corresponding to a crystallographic index of [200], 13.09° corresponding to a crystallographic index of [400], 40.4° corresponding to a crystallographic index of [040], etc. It was found that the 0-3 piezoelectric films showed the main characteristic peak positions of (ATHP)_2_PbBr_4_ powder, indicating that the prepared films did not change the material. Some of the remaining peak positions are known to be characteristic of PVDF, for example at 20.6°, 30°, and 35°. The intensity of the main peak of the 0-3 type films can reach about 90,000, and the higher intensity of the peak indicates a high content of this phase, a more crystalline film, and a large grain size, which corresponds to sequential growth of the crystal plane. This corresponds to previous reports [13] and to the above XRD results for (ATHP)_2_PbBr_4_ powder. Appendix A shows the XRD images of PVDF films at an annealing temperature of 90 °C. PVDF films were heat treated between 65 and 120 °C to obtain β-crystalline films. At this point the film had a relatively strong piezoelectricity with a d_33_ of about 20 pC/N. The d_33_ of the 0-3 type film was known to be 26 pC/N in our subsequent tests. Therefore, the piezoelectricity of the 0-3 type film is provided by both (ATHP)_2_PbBr_4_ and PVDF. In addition, the working temperature of the film should be below 150 °C.

### 3.3. SEM Characterization of 0-3 Type Piezoelectric Films

The accuracy of the material was determined by the above elemental analysis of the prepared films, based on which the annealed completed 0-3 piezoelectric films were characterized and analyzed. Figure 4 shows the 0-3 piezoelectric films annealed at 90 °C for 30 min at 500–1000 rpm. The top view shows that the distribution of the piezoelectric material is relatively uniform, the growth size is also relatively consistent, and all crystals grow in one direction. The thickness of the film is about 5 μm, and the piezoelectric materials are mostly distributed in the middle layer from the cross-section, which indicates that the piezoelectric materials are well dispersed in PVDF.

### 3.4. PFM Piezoelectric Response Study of 0-3 Type Piezoelectric Films

The piezoelectric response of the 0-3 type piezoelectric film was further investigated at an AC drive voltage frequency of 300 kHz, a DC bias voltage of −10 V, and an AC drive voltage of +5 V. Figure 5a–c show the perpendicular piezoelectric response hysteresis curves of 0-3 piezoelectric films. AFM high-resolution morphology maps were obtained using a piezoelectric microscope with a selected scanning area of 2 µm × 2 µm (Figure 5a). A point in the scanning area (blue point in Figure 5a) was selected for the PFM test, and the black curve indicates a change in tip scan bias from −10 V to +10 V, and the red curve indicates a change in tip scan bias from +10 V to −10 V. As shown in Figure 5b,c, it was found that the forward-scan and reverse sweep curves do not overlap. As seen in Figure 5b, the amplitude of the tip DC bias voltage of −10 V is smaller than that of +10 V. Moreover, the minimum point of the amplitude is moving to the left along the x-coordinate axis, and the maximum value of the amplitude is about 0.2 nm. This indicates that the spontaneous polarization direction of the 0-3 piezoelectric film is downward, and the polarization tendency inside the film is also downward in phase. The phase angle in Figure 5c is +115° when the tip bias voltage is −10 V and −65° at −10 V. The difference between the two is 180°, which exactly satisfies the flip angle of the ferroelectric domain and proves that the film is ferroelectric. In addition, the change of phase angle also satisfies the trend of polarization tendency inside the film, both downward, which is consistent with Figure 5b. Since the 0-3 piezoelectric film contains only about 25% of (ATHP)_2_PbBr_4_, and the thickness of the film is thick, the effective (ATHP)_2_PbBr_4_ content is not high, so the piezoelectric response amplitude is not high. The analysis of the phase diagram shows that the ferroelectric domains of the film are flipped nearly 180° and have a downward polarization tendency, indicating that the grain growth has the advantage of meritocratic orientation, which corresponds exactly to the XRD pattern of the previously studied 0-3 piezoelectric film (Figure 3), which is also a basis for the future study of the film polarization.

### 3.5. The 0-3 Type Piezoelectric Film Polarization

In order to improve the piezoelectricity of 0-3 films, they were polarized. According to the experimental conditions and the quality of the sample film, the corona polarization method, which is simple and does not cause much damage to the film, was chosen. Using the previously prepared 0-3 type piezoelectric film sample, the corona polarization did not require electrodes, so the film was not prepared with electrodes. The film was first given upper and lower electrodes to be attached to the conductive substrate, and the unpolarized film was subjected to quasi-static test instrumentation to measure d_33_, and the test results are shown in Table 3. The results show that the d_33_ of the 0-3 film was 2 PC/N, which is consistent with the PFM piezoelectric response test. Next, the 0-3 films were subjected to positive polarization and negative polarization, respectively. Positive polarization was performed by connecting the corona needle to the positive side of the piezoelectric DC voltage and the negative side to the conductive substrate, while negative polarization was exactly the opposite of the positive polarization connection. The polarization condition was that the distance of the corona needle from the film was 1 cm, and the polarization voltage was adjusted to the point where the blue light from the tip of the corona needle could be clearly seen, which means that the air was ionized at this time. If the voltage adjustment is too large, the prolonged polarization can burn the film, but if it is too small, the polarization conditions cannot be met, so this voltage was the appropriate voltage. The voltage adjustment range was finally determined to be between 8.5 KV and 10 KV, and the polarization time was set to 4 h. After polarization, the sample was attached to the electrode, and the d_33_ was measured using a quasi-static test instrument (Table 3). The d_33_ of the film became 0 PC/N after positive polarization and 26 PC/N after negative polarization. The d_33_ was present before polarization and became 0 PC/N after positive polarization. The ions in piezoelectric film are accelerated by the electric field and adhere to the surface or superficial layer of the piezoelectric film in large quantities, so that the ferroelectric domains in the piezoelectric material are flipped in the opposite direction of the unpolarized ferroelectric domains, so the polarization causes a counteracting effect and makes the piezoelectricity basically leave. The piezoelectricity of the film was greatly enhanced after negative polarization, mainly because the above study of XRD of 0-3 type films revealed that the 0-3 type piezoelectric film sample at 6.5° corresponded to a crystalline surface index of [200] with a high peak intensity of the main peak, and the higher intensity of the peak indicated a high content of the phase, a better crystallinity of the film, and a large grain size, which corresponds to sequential growth of the crystalline surface. According to previous reports [13] and XRD analysis of piezoelectric material (ATHP)_2_PbBr_4_ powder in the supporting material, it is known that the intensity of the main peak [200] is very high, indicating that the film has a tendency to grow meritocratically along the main peak with a crystallographic surface index of [200]. This piezoelectric material exhibits excellent piezoelectric properties only when the crystal growth is oriented, which exactly corresponds to the results of our study. Comparison of our experimental results with literature reports [13] reveals that the d_33_ of pure (ATHP)_2_PbBr_4_ films in the literature is 76 pC/N, which is much larger than our experimental value. This is because the content of (ATHP)_2_PbBr_4_ within the 0-3 films is only about 25%, so the piezoelectricity provided is limited. Even so, the d_33_ value we obtained is higher than a quarter of the literature value, again indicating that the piezoelectric properties of the type 0-3 film are provided by both (ATHP)_2_PbBr_4_ and PVDF.

### 3.6. Dielectric Properties and Piezoelectric Voltage Constants of 0-3 Type Piezoelectric Films

The dielectric properties of the piezoelectric film are reflected in the polarization properties and dielectric properties of the dielectric under the electric field, expressed by ε. The ability to bind the charge is enhanced with the increase of the dielectric constant; the polarization strength of the material will be large and not easily polarized, and, conversely, the dielectric constant is small and easily polarized. The dielectric constant formula is:(1)$$\varepsilon =\frac{Cd}{S},$$
where C is the capacitance value of the sample at 1 kHz frequency; d is the thickness of the piezoelectric film; S is the effective area; the capacitance value is measured using an impedance analyzer, which in turn calculates the dielectric constant to be 7.25 × 10^−11^ F/M.

The piezoelectric voltage constant is the coupling between the elastic and dielectric properties of the piezoelectric material and represents the potential shift relationship generated by the strain or the electric field generated by the internal stress, usually expressed as g_33_. The relationship between the piezoelectric voltage constant and the dielectric constant is shown below:(2)$$g_{33}=\frac{d_{33}}{\varepsilon }$$

From the above study, it is known that the d_33_ of the material is 26 PC/N, which is brought in to calculate g_33_ =358.6 × 10^−3^ Vm/N. According to the literature [13], it is known that the maximum value of g_33_ of PVDF is 286.7 × 10^−3^ Vm/N, which shows that the g_33_ of 0-3 piezoelectric film is still relatively large, and the large piezoelectric voltage constant helps to improve the output of the sensor, which is important for the manufacture of high-performance sensors to have important significance.

## 4. Conclusions

In this paper, the ferroelectric and piezoelectric properties of 0-3 piezoelectric films based on (ATHP)_2_PbBr_4_ piezoelectric material were investigated, and the crystal structure and phase transition temperature of (ATHP)_2_PbBr_4_ were determined by XRD and DSC analysis. The temperature of (ATHP)_2_PbBr_4_ powder ferroelectric phase transition was measured to be about 220 °C. Next, 0-3 piezoelectric films were prepared using the homogenization method, and the piezoelectric films were characterized and analyzed using XRD, SEM, and EDS. The structural characteristics of the films were analyzed by XRD. The elemental composition of the piezoelectric films was determined by EDS characterization, and the distribution of the piezoelectric materials, (ATHP)_2_PbBr_4_ and PVDF, was indirectly determined. The 0-3 piezoelectric films were spot tested using PFM, and the films were found to have good ferroelectricity and piezoelectricity throughout the study, and the piezoelectric films had a downward polarization tendency. After corona polarization of the films, it was determined that the piezoelectricity of the films could be improved by negative polarization for 0-3 piezoelectric films, and the effect of polarization on the piezoelectric properties of (ATHP)_2_PbBr_4_-based piezoelectric films was carefully analyzed. Finally, the piezoelectric voltage constant g_33_=358.6 × 10^−3^ Vm/N of the piezoelectric film was calculated to be higher than that of the PVDF material, which is important for the fabrication of high-performance sensors.

## Figures and Tables

**Figure 1 materials-15-06378-f001:**
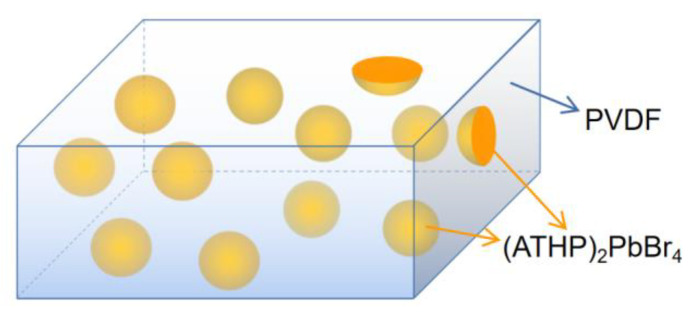
The 0-3 type composite piezoelectric film structure schematic.

**Figure 2 materials-15-06378-f002:**
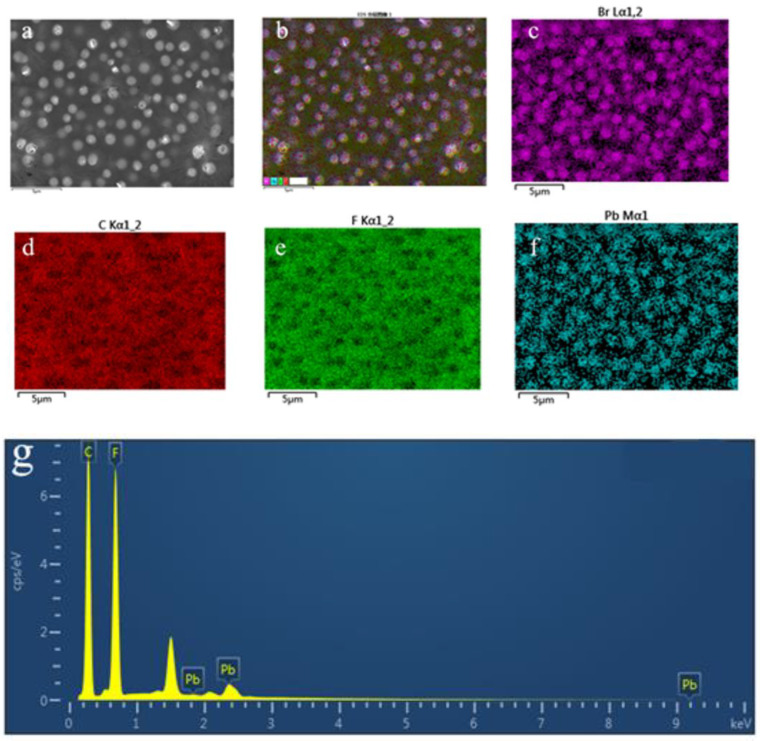
The 0-3 type piezoelectric thin film X-ray energy spectrum analysis: (**a**) energy spectrum surface scan area, (**b**–**f**) EDS element distribution map, (**g**) X-ray energy spectrum.

**Figure 3 materials-15-06378-f003:**
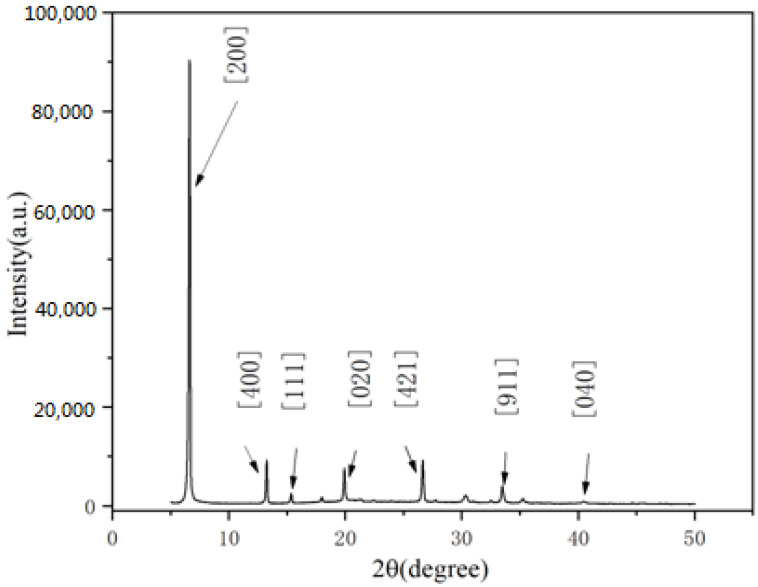
XRD pattern of 0-3 piezoelectric film at an annealing temperature of 90 °C at a speed of 500–1000 rpm.

**Figure 4 materials-15-06378-f004:**
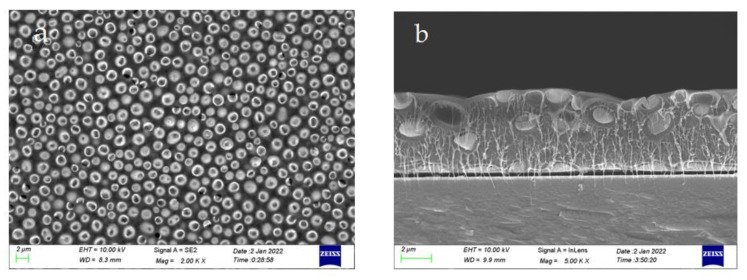
The 0-3 type piezoelectric film morphology at 500–1000rpm speed ((**a**,**b**) annealing temperature 90 °C, 20,000× top view, 50,000× cross-sectional view).

**Figure 5 materials-15-06378-f005:**
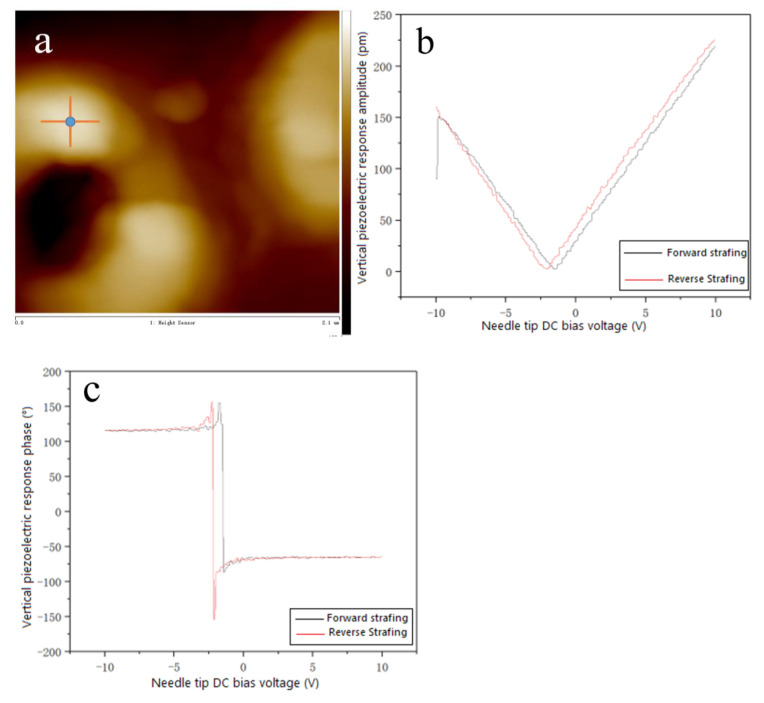
Vertical piezoelectric response hysteresis curve based on (ATHP)_2_PbBr_4_ piezoelectric film. (**a**) high-resolution topography, (**b**) piezoelectric response amplitude hysteresis curve in the vertical direction, (**c**) piezoelectric response phase hysteresis curve in the vertical direction.

**Table 1 materials-15-06378-t001:** Experimental use of material.

Drug Names	Molecular Formula	Purity	Manufacturer
De-ionized water	H_2_O	Electrical conductivity 0.0055 μs/cm	ShuyangXizhimeng Co. from Jiangsu, China
Anhydrous ethanol	CH_3_CH_2_OH	AR ^1^	Sinopharm Group Chemical Reagent Co. from Shanghai, China
Isopropyl alcohol	(CH_3_)_2_CHOH	AR ^1^	Sinopharm Group Chemical Reagent Co.
Dimethylformamide anhydrous (DMF)	C_3_H_7_NO	99.8%	Alfa Aesar (China) Chemical Co. from Shanghai, China
Acetone	CH_3_COCH_3_	AR ^1^	Sinopharm Group Chemical Reagent Co.
Lead bromide	PbBr_2_	99.00%	Shanghai Maclean Biochemical Technology Co. from Shanghai, China
4-Aminotetrahydropyran	C_5_H_12_NO	97.00%	Shanghai Maclean Biochemical Technology Co.
Polyvinylidene fluoride	(CH_2_CF_2_)n	average Mw~400,000, powder	Shanghai Maclean Biochemical Technology Co.
Hydrobromic acid	HBr	AR ^1^	Sinopharm Group Chemical Reagent Co.

^1^ AR is analytical purity.

**Table 2 materials-15-06378-t002:** Selected element types and content percentages obtained by X-ray spectroscopy of 0-3 piezoelectric films.

Element Symbol	Element Name	Weight Percentage	Atomic Percentage
C	carbon	48.95	68.86
F	fluorine	31.56	28.07
Pb	lead	8.04	0.66
Br	bromine	11.45	2.42
total		100.00	100.00

**Table 3 materials-15-06378-t003:** Comparison of the piezoelectric properties of 0-3 type films after different polarization processes.

	Unpolarized (d_33_)	Anodization (d_33_)	Negative Polarization(d_33_)
0-3 type piezoelectric film	2 PC/N	0 PC/N	26 PC/N

## Data Availability

The data used to support the findings of this study are available from the corresponding author upon request.

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
