# Peer review of "Piezoelectric Properties of 0-3 Composite Films Based on Novel Molecular Piezoelectric Material (ATHP)2PbBr4"

_materials, 2022, doi:10.3390/ma15186378_

Round 1

Author Response

We are grateful to the reviewers for their positive comments and constructive suggestions on our work, which have helped us to further improve the manuscript. We have adequately addressed these comments in the revised manuscript. Revised sentences and words have been marked with the revision function to make them easier to find in the manuscript. Below is our point-by-point response to the reviewers' comments, and our responses are shown in red font.

Reviewer 2 Report

The manuscript entails an interesting topic where piezoelectric properties of PVDF/ (ATHP)2PbBr4 is discussed. However, the manuscript with the current form lacks some major issues and cannot be accepted for publication as it is. The manuscript needs serious revisions, some of which are mentioned below.

1)     Title needs to be reformed. Excessive use of “piezoelectric” in one sentence. Plus, it does not make sense to say, “Preparation and piezoelectricity…”. probably “piezoelectric properties” would be proper term to use.

2)     KNN? what is KNN? Please use the full name in first occasion.

3)     What is the novelty of this work? It must be clearly addressed in the introduction with respect to its difference with previous studies.

4)     Section 2: Main materials and testing instruments should be included here not in a supporting document and in a proper way. Only mentioning the manufacturer is not enough. Testing methods must be declared clearly. for example, temperature ranges observed in DSC, flow rates, wavelength in XRD, 2 theta angle ranges and many more.

5)     Section 2: drug? It is recommended to use "material" instead of drug.

6)     Section 2.1: This section is unnecessarily too long. Please make it concise and include only the vital parts.

7)     What is the concentration of the (ATHP)2PbBr4 powder in the final composite film?

8)     Result and Discussion: Figure S3 is not discussed at all. There must be an explanation on XRD pattern of the composite films. Probably formation of β phase should be discussed with respect to the peak positions. This must be compared with XRD patterns of the neat PVDF.

9)     Figure S2: What about the DSC of the composite films? As mentioned earlier the conditions of measurements must be stated clearly.

10)  Figure 3: Why c and d? are there any a and b images?

11)  Figure 4: again, why d-f? aren't there a-c figures? Quality of the figures are poor. inset legends are not easily visible.  What does Figure 4d represent?

12)  Page 6 line 194-197: when was this discussed?

13)  Figure 4: caption must be clear and explain each individual figure.

14)  Section 3.4: “To improve the piezoelectricity of piezoelectric films.” Piezoelectricity of the films.

15)  Page 7 line 230 to 233: “…0-3 piezoelectric thin film sample at 6.5° corresponds to a crystallographic surface index of [200] with a high peak intensity of the main peak…” Please refer to the figure.

16)  In fact, FWHM is more accurate analytic parameter to discuss about the grain size than the intensity. you may need to use Scherrer formula to confirm this.

17)  Section 3.5: It is recommended to present the dielectric constant curves with respect to the frequency ranges and temperature of intention.

18)  Page 7: “According to the literature [6], it is known…” Is this correct? Please double check. I could not find this value in this citation. Probably different citation should be referred.

19)  References: some of the references are referred incorrectly in terms of their date. For example [6] and [11].

20)  Please use the proper citation for [7].

21)  The number of citations is relatively low. I am positive there are more studies on the piezoelectric behaviour of PVDF and polymer composites.

In general, I believe the results are poorly presented and are inadequate. For example, formation of β phase is quite important in piezoelectric and dielectric properties of PVDF, but it has not been discussed at all either in XRD results nor other methods. FTIR is another tool that could be used to detect this property. But it has not been investigated. Dielectric properties should be presented in a proper way via plots against frequency ranges and temperatures. And they must be compared with those of the neat PVDF to show enhancements.

Experimental section suffers from poor explanations. Each instrument must be explained properly with respect to their conditions.

Author Response

(The authors gave the same response as above.)

Round 2

Reviewer 2 Report

Considering the provided responses to the previously raised concerns, I am convinced that the authors have kindly addressed the comments and the manuscript now could be accepted for publication.